# Short communication: Driftwood provides reliable chronological markers in Arctic coastal deposits

Lasse Sander[1], Alexander Kirdyanov[2,3], Alan Crivellaro[4,5], Ulf Büntgen[4,6,7,8]

[1] Alfred Wegener Institute, Helmholtz Centre for Polar and Marine Research, List/Sylt, Germany.
[2] V.N. Sukachev Institute of Forest SB RAS, Federal Research Centre, Krasnoyarsk, Russia.
[3] Institute of Ecology and Geography, Siberian Federal University, Krasnoyarsk, 660041, Russia.
[4] Department of Geography, University of Cambridge, Cambridge, United Kingdom.
[5] "Stefan cel Mare" University of Suceava. Suceava, Romania.
[6] Swiss Federal Research Institute WSL, CH-8903 Birmensdorf, Switzerland.
[7] Global Change Research Institute CAS, 603 00 Brno, Czech Republic.
[8] Faculty of Science, Department of Geography, Masaryk University, 611 37 Brno, Czech Republic.

*Correspondence to*: Lasse Sander (lasse.sander@awi.de)

**Abstract:**

Originating from the boreal forest and often transported over large distances, driftwood characterises many Arctic coastlines. Here we present a combined assessment of radiocarbon ($^{14}$C) and dendrochronological (ring width) age estimates of driftwood samples to constrain the progradation of two Holocene beach-ridge systems near the Lena Delta in the Siberian Arctic (Laptev Sea). Our data show that the $^{14}$C ages obtained on syndepositional driftwood from beach deposits yield surprisingly coherent chronologies for the coastal evolution of the field sites. The dendrochronological analysis of wood from modern driftlines revealed the origin and recent delivery of the wood from the Lena River catchments. This finding suggests that the duration of transport lies within the uncertainty of state-of-the-art $^{14}$C dating and thus substantiates the validity of age indication obtained from driftwood. This observation will help to better understand changes in similar coastal environments, and to improve our knowledge about the response of coastal systems to past climate and sea-level changes.

**Keywords:** Arctic driftwood; beach deposits; Holocene; dendrochronology; radiocarbon dating

# 1 Introduction

The Arctic Ocean is strongly influenced by terrestrial runoff, and driftwood is a common feature in its coastal waters (Peterson et al., 2002; Yang et al., 2002; Woo and Thorne 2003). Arctic driftwood mainly originates from the largest rivers in the boreal forests of Eurasia and North America, and trees are primarily mobilized by river bank erosion during peak summer runoff (Gurnell et al., 2002; Costard et al., 2014; Kramer & Wohl, 2016). Studies in Arctic coastal and marine environments have used the presence, location, species, and age of driftwood to infer the long-term variability of ocean current dynamics, sea-ice conditions, and the occurrence of storm surges (e.g. Reimnitz and Maurer, 1979; Harper et al., 1988; Dyke et al., 1997; Tremblay et al. 1997; Bennike, 2004; Polyak et al., 2010; Nixon et al., 2016; Irrgang et al., 2019).

Age control on Holocene driftwood is commonly obtained by radiocarbon ($^{14}$C) dating and the reported uncalibrated ages may have uncertainties in the order of decades to centuries (e.g. Funder et al., 2010; Hole and Macias-Fauria, 2017). Modern AMS facilities, however, can provide substantially lower methodological dating uncertainties (Wacker et al., 2010; Nixon et al., 2016; Mason et al., 2020; Fig. S1). Radiocarbon dated driftwood has also been used to indirectly infer the age of wave-built coastal deposits, alongside other organic materials, such as human artefacts, whale bones, or mollusk shells (e.g. Dyke et al., 1991; Forman et al., 1996; Forman et al., 2007; Funder et al., 2011). The advantage of using a terrestrial deposit for radiocarbon dating in a marine context is that reservoir corrections are not necessary. This is especially useful in brackish coastal environments in the proximity to major rivers and other sources of old carbon (Forman and Polyak, 1997; Dumond and Griffin 2002; Grigoriev et al., 2004). Alternatively, well-preserved driftwood can be dated by cross-correlating their tree-ring width measurements against independent, species-specific reference chronologies from the boreal catchments (Hellmann et al., 2013; 2017), often yielding indication on the age of the outermost tree ring with a resolution of years to decades (Eggertson, 1994; Johansen, 1998; Steelandt et al., 2015).

A large unknown conceptual uncertainty lies in the pathway and duration of wood transport in the riverine and marine environments prior to deposition in a coastal system. Wood transport in rivers is not particularly well-investigated and depends largely on catchment properties and runoff regime (Kramer and Wohl, 2016; Kramer et al., 2017). The few studies available suggest that residence times in

watersheds are in the order of several years to centuries, with large uncertainties on the fate of the wood following recruitment (Swanson & Lienkaemper, 1978; Schenk et al., 2014; Alix, 2016). Once the driftwood has reached the ocean, it is often argued that the sea-ice-bound transport across the Arctic occurs within a few years and that free-floating wood sinks over a matter of months (Häggblom, 1982;

Dyke et al., 1997; Bennike, 2004; Funder et al., 2011). A relatively rapid dispersal of driftwood is likewise supported by the results of Hellmann et al. (2016a), who studied logging debris from the coasts of Svalbard, Greenland and Iceland by dendrochronological means. The authors showed that the age of the logs correlates well with a historically documented period of intensive forest use in western Siberia in the mid-20th century.

Here, we evaluate the reliability of age estimates from driftwood in an Arctic coastal environment near a large forested source area by combining the $^{14}$C dating of buried Holocene wood from prograded beach deposits with the dendrochronological cross-dating of modern driftwood samples from the same coastal site. Obtaining reliable age-control with a sufficiently high resolution is crucial to identify small changes in the long-term rates of past coastal evolution in relation to external effects of climate, sea level, or storm

surges.

## 2 Materials and methods

### 2.1 Research area

Driftwood samples from two field sites along the south-western seaboard of Buor Khaya Bay (Laptev Sea, Sakha Republic, Russian Federation) are investigated. The northern site is referred to as Bys Tasa

(BY; 71°2'N; 130°11'E) and the southern site as Makhchar (MA; 70°59'N, 130°14'E). Due to the proximity to the Lena Delta (Büntgen et al., 2014), the brackish surface water conditions in Buor Khaya Bay during the ice-free summer months contain large quantities of driftwood (Pavlov et al., 1996; Günther et al., 2011). The area is microtidal (tidal range: ~0.4 m), but water levels are subject to considerable wind-forced variation with surge heights of up to 2.5 m above mean sea level (Ashik and Vanda, 1995;

Pavlov et al., 1996; Lantuit et al., 2012). BY and MA are comprised by wide sets of coarse-clastic beach ridges that separate coastal lagoons from the open water of the bay. Beach ridges form at locations along

wave-dominated shorelines, where accommodation and sediment supply have been positive over extensive periods of time. Prograded beach-ridge systems and spits are recurrent features along the shoreline of the eastern Laptev Sea, but remain largely unstudied (with the exception of Sander et al., 2019). In most cases, each visible ridge is a time-integrated feature formed during multiple periods of reworking rather than during an individual event (Tamura, 2012; Lindhorst and Schutter, 2014). At the two studied field sites, driftwood is present as syndepositional debris within the beach deposits, at different elevations on the modern beach surface, and as overtopped debris (Fig. 1, Top right).


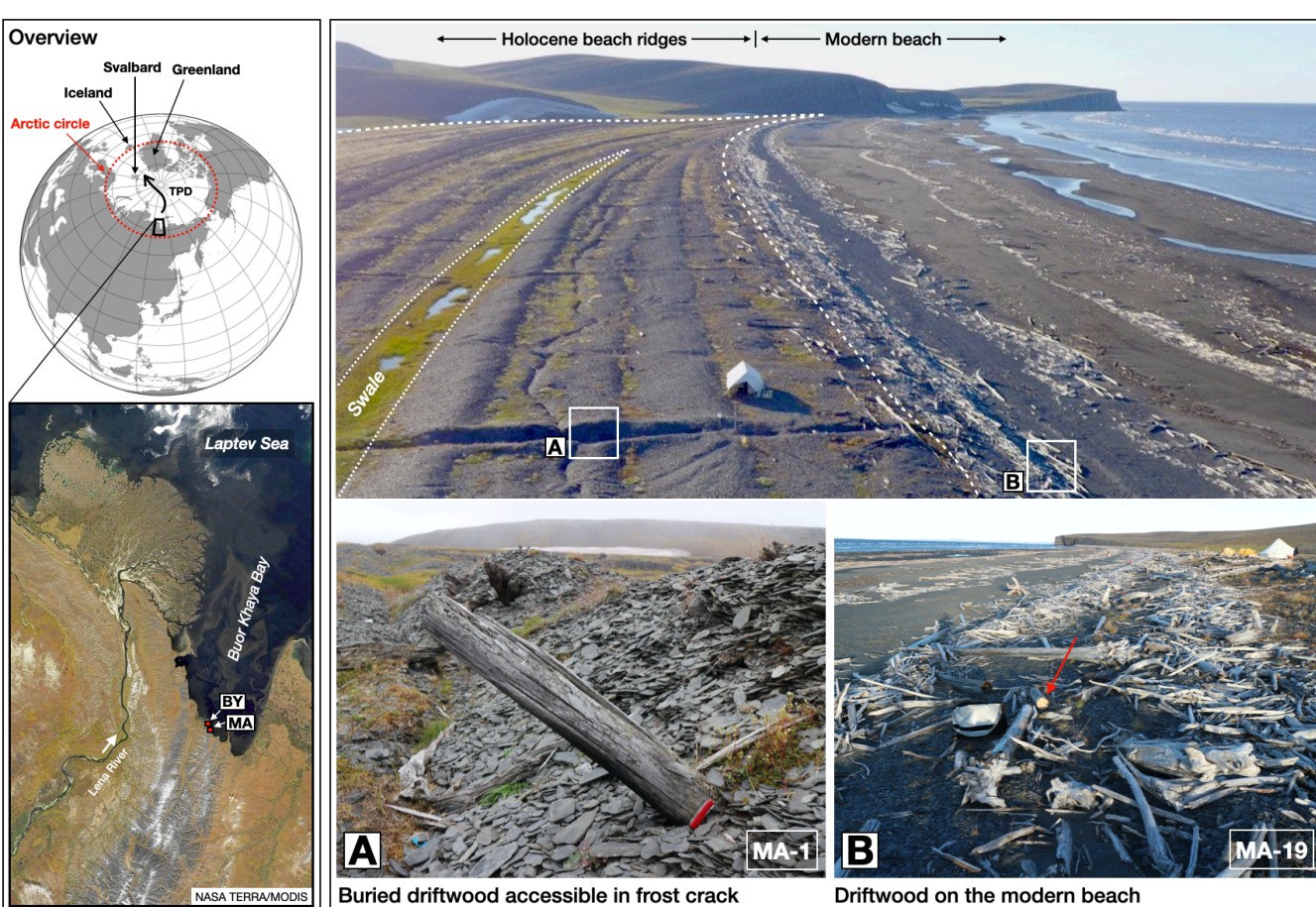

**Figure 1: Left: Overview sketch of the location of the research area, Top right: Aerial view of the Makhchar field site showing the location of the Holocene beach ridges and the modern beach with the distribution of wood arranged in driftlines from high-, medium-, and low-energy situation,**

**Bottom right: (A) Buried Holocene driftwood accessible in a frost crack, and (B) a dense assemblage of driftwood on the modern beach face (code indicates sample ID, cf. Table 2).**

## 2.2 Sampling design

A total of 28 driftwood samples were taken for radiocarbon dating to establish age control on the beach-ridge sequences at BY and MA. The driftwood was accessible in frost cracks and in areas of intensive cryoturbation (Fig. 1A). Where possible, multiple samples of distinct driftwood were collected from each location (here referred to as sets; Table 1 & 2). The wood was stabilized in situ with string/duct tape to prevent disintegration. Collected with a manually operated saw, the samples were stored in the field under well-aerated conditions and later cleaned in the laboratory and dried for at least 48 hours at 45°C. For stem disks, subsamples for $^{14}$C dating were taken from the outermost preserved tree rings and submitted to the AMS Laboratory in Bremerhaven, Germany (i.e. the MICADAS at Alfred Wegener Institute). A description of the sample preparation and dating protocol can be found in Sander et al. (2019). The obtained $^{14}$C ages were calibrated using OxCal 4.3 (Bronk Ramsey, 2009) and the IntCal13 calibration curve (Reimer et al., 2013). All ages are reported in the BCE/CE system to allow a better comparison with the modern driftwood data. For the two beach-ridge systems, Bayesian age-distance models (P-sequences, Fig. S2&S3) were constructed in OxCal 4.3 using the distance of each sample from the innermost preserved beach ridge as a spatial and chronological reference (Bronk Ramsey, 2008; Sander et al., 2019).

At MA, 26 stem disks were taken from driftwood along the modern beach for dendrochronological age-determination and wood anatomical species-identification. The analyses were conducted at the Tree-Ring Unit at the Department of Geography, University of Cambridge, UK. The surface of the discs was polished with sand paper of progressively fine grain size to 600 grid and tree-ring widths of all samples were measured on a Velmex Tree Ring Measuring System with a resolution of 0.001mm (Velmex Inc., Bloomfield, NY, USA). The individual series were cross-dated against regional reference chronologies from northern Yakutia (Hellmann et al. 2016b) using TSAP-win (Rinntech, Heidelberg, Germany) and COFECHA (Version 6.02P). Wood anatomical sections were cut with a razor blade from transverse,

radial and tangential sections, and mounted with water between two cover glasses and analysed using an Olympus CX22LED microscope at 20-400x magnification. Based on wood macroscopic (Ruffinatto et al., 2019) and anatomical (Wheeler et al., 1989, Richter et al., 2004) features, each sample was identified

125 to the species or closest possible taxon level using reference identification keys (Benkova and Schweingruber, 2004).

**Table 1: Overview of radiocarbon samples from Bys Tasa (BY)**

| ID | Set | Distance* | Laboratory code | 14C age | Calibrated age (Probability, in %) | Age span | Median age | Genus |
|---|---|---|---|---|---|---|---|---|
| BY-1 | 1 | 0 | 2292.1.1 | 5425±30 | 4341 – 4239 BCE (95.4) | 103 | 4290 BCE | Populus (Salix) |
| BY-2 | 1 | 0 | 2293.1.1 | 5441±29 | 4346 – 4247 BCE (95.4) | 100 | 4293 BCE | Larix (Picea) |
| BY-6 | 2 | 130 | 2297.1.1 | 5063±29 | 3954 – 3792 BCE (95.4) | 163 | 3870 BCE | Picea (Larix) |
| BY-5 | 2 | 140 | 2296.1.1 | 5118±29 | 3978 – 3911 BCE (46.3) 3878 – 3803 BCE (49.1) | 68 76 | 3881 BCE | Larix (Picea) |
| BY-7 | - | 240 | 2298.1.1 | 4697±29 | 3629 – 3586 BCE (14.0) 3531 – 3488 BCE (21.8) 3472 – 3372 BCE (59.7) | 44 44 101 | 3502 BCE | Populus (Salix) |
| BY-3 | 3 | 400 | 2294.1.1 | 4387±29 | 3091 – 2917 BCE (95.4) | 175 | 2990 BCE | Larix (Picea) |
| BY-4 | 3 | 400 | 2295.1.1 | 4380±29 | 3090 – 3046 BCE (14.6) 3036 – 2913 BCE (80.8) | 44 123 | 2984 BCE | Pinus |
| K3** | - | 510 | 1290.2.2 | 4038±49 | 2855 – 2812 BCE (7.2) 2747 – 2725 BCE (1.9) 2698 – 2466 BCE (86.3) | 43 22 233 | 2579 BCE | - |
| K4** | - | 590 | 1291.2.2 | 5441±29 | 2479 – 2206 BCE (95.4) | 273 | 2311 BCE | - |
| K2** | 4 | 700 | 1289.2.2 | 3500±48 | 1945 – 1729 BCE (89.9) 1723 – 1692 BCE (5.5) | 216 31 | 1823 BCE | - |
| BY-8 | 4 | 700 | 2299.1.1 | 3544±28 | 1955 – 1771 BCE (95.4) | 184 | 1889 BCE | Picea (Larix) |
| K1** | - | 880 | 1288.2.2 | 1288±48 | 1612 – 1404 BCE (95.4) | 208 | 1422 BCE | - |
| BY-9 | - | 1100 | 2300.1.1 | 1148±27 | 776 –794 CE (7.7) 800 – 973 CE (87.7) | 18 174 | 894 CE | Larix |

*Distance from the innermost beach ridge, in m
130 ** Sampled in 2017

**Table 2: Overview of radiocarbon samples from Makhchar (MA)**

| ID | Set | Distance * | Laboratory code | 14C age | Calibrated age (Probability, in %) | Age span | Median age | Genus |
|---|---|---|---|---|---|---|---|---|
| MA-41 | 5 | 0 | 2290.1.1 | 5294±29 | 4235 – 4041 BCE (95.0)<br>4008 – 4006 BCE (0.4) | 195<br>2 | 4132 BCE | Larix |
| MA-42 | 5 | 0 | 2291.1.1 | 5153±29 | 4041 – 4013 BCE (9.4)<br>4002 – 3938 BCE (76.7)<br>3860 – 3814 BCE (9.2) | 28<br>65<br>47 | 3968 BCE | Picea (Larix) |
| MA-40 | 5 | 0 | 2289.1.1 | 5127±29 | 3986 – 3914 BCE (55.1)<br>3878 – 3804 BCE (40.3) | 73<br>75 | 3942 BCE | Larix (Picea) |
| MA-32 | 6 | 280 | 2286.1.1 | 4394±28 | 3091 – 2921 BCE (95.4) | 171 | 3000 BCE | Populus (Salix) |
| MA-33 | 6 | 280 | 2287.1.1 | 4322±29 | 3016 – 2891 BCE (95.4) | 85 | 2925 BCE | Picea |
| MA-30 | 7 | 315 | 2284.1.1 | 4382±29 | 3090 – 3046 BCE (15.7)<br>3036 – 2914 BCE (79.7) | 45<br>123 | 2986 BCE | Picea |
| MA-31 | 7 | 315 | 2285.1.1 | 4344±29 | 3078 – 3074 BCE (0.7)<br>3024 – 2899 BCE (94.7) | 5<br>126 | 2960 BCE | Abies |
| MA-27 | 8 | 410 | 2281.1.1 | 4149±28 | 2875 – 2830 BCE (18.6)<br>2822 – 2630 BCE (76.8) | 46<br>193 | 2744 BCE | Picea (Larix) |
| MA-28 | 8 | 410 | 2282.1.1 | 4169±28 | 2881 – 2833 BCE (19.8)<br>2819 – 2662 BCE (73.7)<br>2649 – 2636 BCE (1.9) | 49<br>158<br>14 | 2765 BCE | Populus (Salix) |
| MA-29 | 8 | 410 | 2283.1.1 | 4175±28 | 2883 – 2835 BCE (20.7)<br>2817 – 2666 BCE (74.7) | 49<br>152 | 2771 BCE | Larix (Picea) |
| MA-39 | - | 860 | 2288.1.1 | 2121±28 | 343 – 325 BCE (3.3)<br>205 –52 BCE (92.1) | 19<br>154 | 146 BCE | Picea (Larix) |
| MA-4 | 9 | 915 | 2279.1.1 | 1133±27 | 777 – 791 CE (3.1)<br>807 – 842 CE (4.8)<br>861 – 986 CE (87.4) | 14<br>35<br>126 | 924 CE | Picea (Larix) |
| MA-3 | 9 | 940 | 2278.1.1 | 944±27 | 1027 – 1155 CE (95.4) | 129 | 1098 CE | Larix (Picea) |
| MA-1 | 9 | 965 | 2277.1.1 | 770±27 | 1219 – 1280 CE (95.4) | 61 | 1255 CE | Larix (Picea) |
| MA-26 | 9 | 970 | 2280.1.1 | 568±27 | 1307 – 1363 CE (56.2)<br>1385 – 1423 CE (39.2) | 56<br>38 | 1350 CE | Larix (Picea) |

*Distance from the innermost beach ridge, in m

# 3 Results and discussion

## 3.1 Holocene driftwood

The 13 samples from BY yielded ages between 4300 BCE and 900 CE, and the 15 samples from MA exhibited ages between 4100 BCE and 1400 CE. These dates establish a robust mid- to late Holocene formation of the two beach ridge systems. The four sets sampled at BY yielded uncalibrated [14]C ages with overlapping uncertainties, hence resulting in very similar calibrated age ranges (Fig. 2A; Table 1, sets 1–4). The analysis of wood anatomy for the densely spaced sets revealed, that different genera were dated and rules out the possibility that parts of the same tree were accidentally sampled (Table 1). At MA, a set was sampled on the innermost and hence oldest preserved beach ridge (set 5, n=3, Table 2). Two of the three samples have very similar median ages (3940 BCE, MA-40; 3970 BCE, MA-42), while one sample is ~200 years older (4130 BCE, MA-41), and the calibrated probabilities (2-sigma) are not overlapping (Fig. 3B). In the lateral part of the beach-ridge system, the ages of three sets, located at cross-ridge distances of 35 m and 95 m, were determined (set 6–8). Similar to BY, the age control within each set is coherent, but the ages of the more closely-spaced sets overlap entirely. However, since both sets are located on either side of the same swale (a morphological depression between beach ridges; cf. Fig. 2), it is not unlikely that both sets, despite their slight spatial offset, originate from the same depositional event (or series of events). All median ages of the two sets lie within a window of 75 years. The [14]C ages of the third set (set 8, n =3) from the lateral part of MA are overlapping entirely. They have the same age within the uncertainty of the method, but do not overlap with sets 6–7 (Fig. 3C). Four samples were taken as a series across two distinct beach ridges in the outermost part of MA. The distance between the inner and the outermost sample is ~50 m. Despite the proximity of the samples, there is no overlap between the 2-sigma age ranges and no age inversions (Fig. 3D, Table 2). The wood anatomical assessment of the MA material confirms different genera were dated within closely-spaced sets.

The driftwood [14]C ages are incorporated into Bayesian age models for beach-ridge progradation (Fig. 3). The rationale behind this is that beach ridges are aligned to the angle of approach of wave energy during formative events. This determines that the innermost beach ridge is oldest, and that the age of each beach ridge is the same along its entire length within the time necessary for beach-ridge formation. This age-distance relationship is assumed to be most pronounced in rapidly prograding systems characterized by

continuous sediment and energy supply, and deposition within a protected environment, hence minimizing reworking or erosion.

165   A comparison of the age models from the two field sites shows a high degree of coherence in the timing and rate of coastal evolution. Both BY and MA initiated around 4000 BCE and prograded over the mid-Holocene at a rate of 0.3 m/yr. Deposition at BY ceased at around 400 BCE, while MA enters a period of slow or punctuated progradation between 1500 BCE and 700 CE, before experiencing renewed progradation between 700 CE and 1400 CE. No beach sediment is preserved at MA after that time. At

170   BY, the mode of beach ridge progradation changed after 400 BCE, and appears to become limited by sediment supply or a limitation in transport capacity (Sander et al., 2019). A single sample (BY-9) from the outer beach ridges is coherent with the period of renewed progradation at MA. The timing of changes in coastal evolution fits reasonably-well with studies proposing the existence of periods of warmer climate conditions in the wider Laptev Sea area during the mid- and late-Holocene (Andreev and Klimanov, 2000;

175   Andreev et al., 2011). However, the current understanding of the timing and magnitude of past climate variability in Siberia remains inconclusive (Büntgen et al. 2014). Furthermore, there is significant offset in previous findings (see Biskaborn et al., 2016 for a review). Our data suggest that prograded coastal systems may provide viable candidates to study the effective importance of variability in wave-climate interaction, as a function of reduced/increased fetch resulting from past changes in the extent and duration

180   of sea-ice cover.

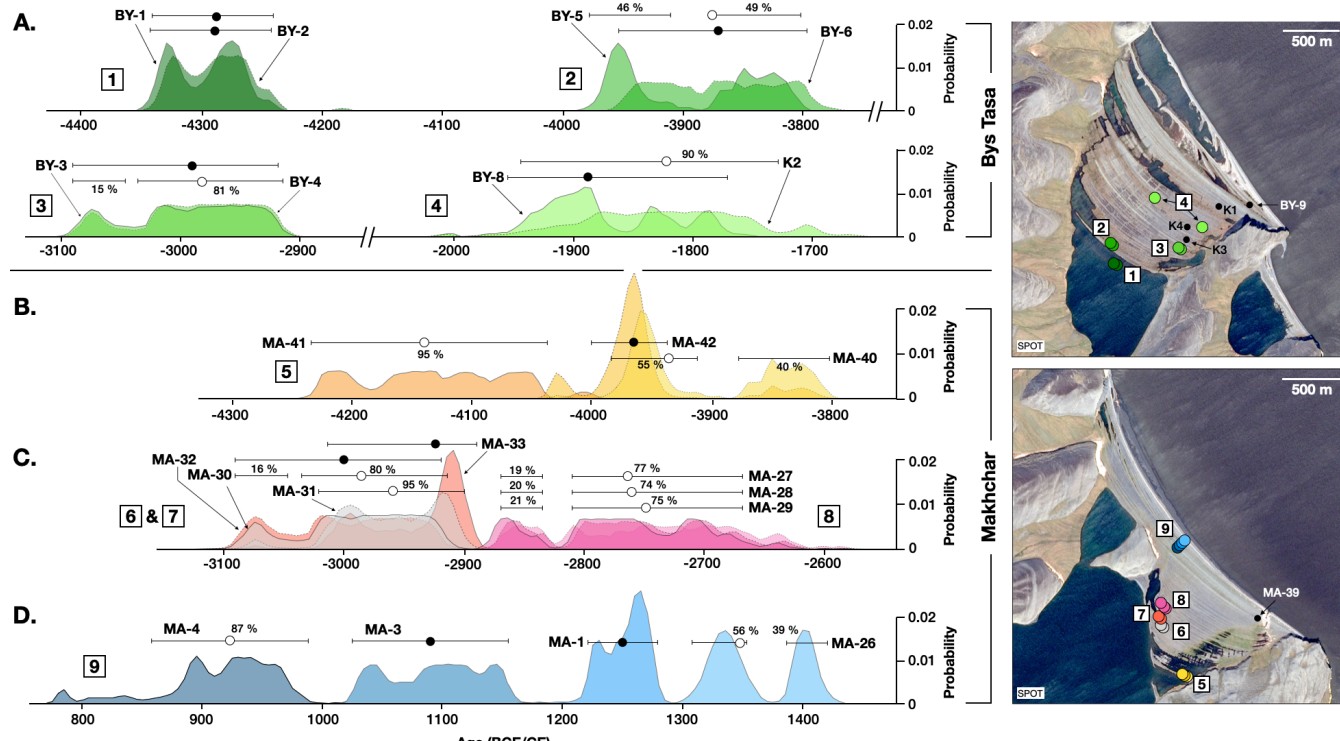

**Figure 2: Comparison of the calibrated radiocarbon probabilities between the sample sets obtained from the field sites.** Two sigma age ranges (95.4%) are indicated by the horizontal bar and the location of the circle indicates the median age of the sample. Empty circles indicate probabilities < 95.4% and the values are given above the respective bar. Probabilities < 10% are not displayed. The figure illustrates that most of the age indications obtained from samples at stratigraphically similar positions overlap almost entirely (A, C), and that sensitive changes in coastal progradation can be captured by closely-spaced samples (D).

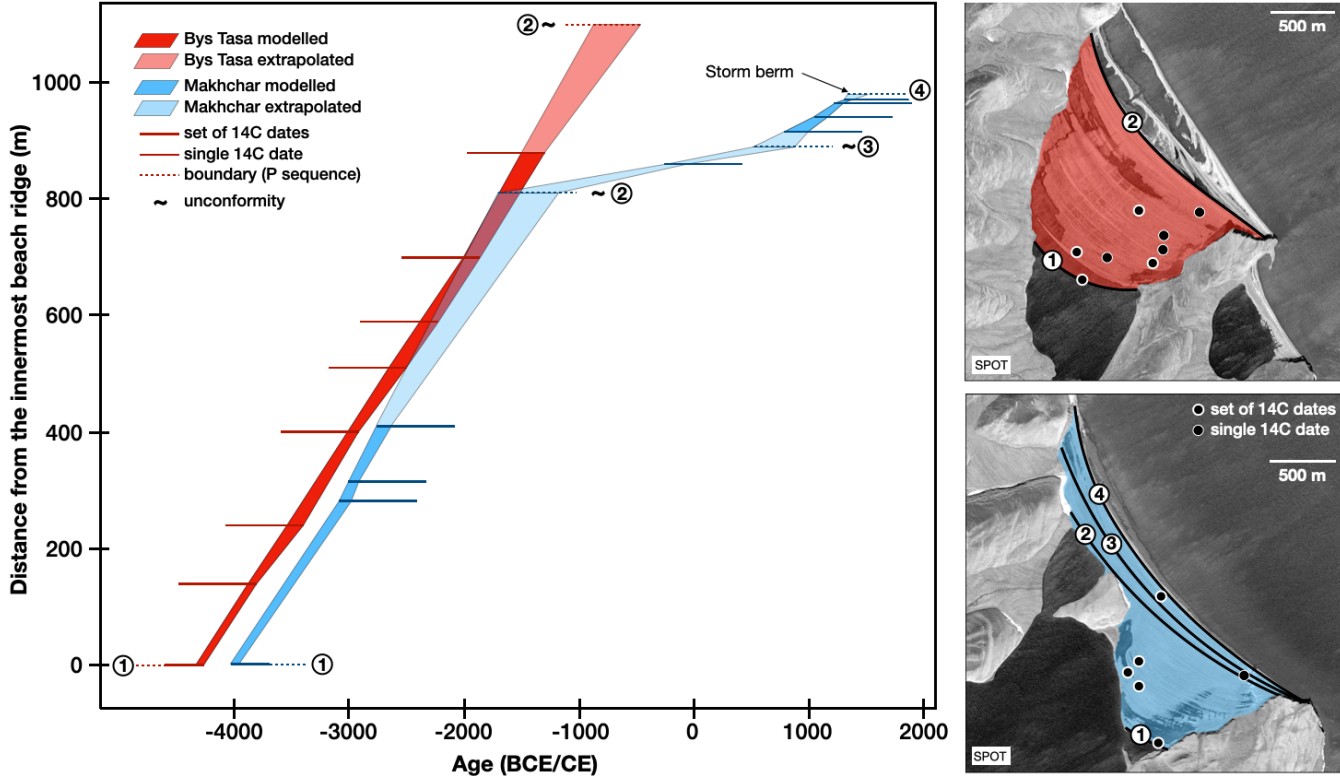

**Figure 3: Age-distance models for the progradation of the two beach-ridge systems. Solid lines mark the position of samples and sample sets. The circled numbers on the right indicate the location of boundaries in the P-sequence, as the first and last preserved ridges and unconformities in the pattern of progradation. The black circles indicate the location of individual samples and sample sets (see Figure 2 and Table 1 & 2 for further details).**

## 3.2 Modern driftwood

Four *Larix* samples from the driftlines of MA were dendro dated and provenanced (Table 3). The youngest tree rings in two samples have calendar ages of 1951 CE and 1954 CE (MA-19, MA-20). The outer rings in two other samples were dated to 1988 CE (MA-34, MA-38). The relatively recent ages of wood samples on the modern beach of Buor Khaya Bay suggest the driftwood has passed its "lifecycle" from erosion to transport and deposition within the last 30 and 70 years, respectively. This agrees with Kramer and Wohl (2016), who propose that the transfer of wood in large rivers (watersheds $>10^6$ km$^2$) can be rapid. This means in practical terms, that the uncertainties associated with the unassessed transport pathways of

driftwood through the Lena catchment are lower than the methodological uncertainties associated with state-of-the-art $^{14}$C dating (Fig. S1). In our case, the number of tree rings in the modern samples contain up to ~300 years, which illustrates that considerable offsets in the quality of the $^{14}$C age determination will occur if the position of a $^{14}$C sample is unknown within the tree-ring sequence. Furthermore,

driftwood may be subject to abrasion during transport or decay after deposition from the exposure to e.g. humidity and biological activity (Fig. S4). The age of most samples (n =21) could not be determined, as the age span of the samples was too short for reliable cross-dating against boreal reference chronologies from north-eastern Siberia (Table 3). However, the emerging possibilities of high-resolution $^{14}$C measurements (Brehm et al. 2021) offer a way to trace the atmospheric $^{14}$C bomb peak of the 1960s (Levin

& Hesshaimer 2000; Andreu-Hayles et al. 2015), and cosmic events (Büntgen et al. 2018), thus represent an additional tool to support the dendro dating of driftwood.

**Table 3: Samples from the modern beach (Makhchar, MA)**

| ID | First year | Last year | Age span* | Reference chronology | Genus |
|---|---|---|---|---|---|
| MA-5 | ? | ? | ~65 | N/A | Picea |
| MA-6 | ? | ? | ~80 | N/A | Picea |
| MA-7 | ? | ? | - | N/A | Picea |
| MA-8 | ? | ? | - | N/A | Alnus |
| MA-10 | ? | ? | ~90 | N/A | Larix |
| MA-11 | ? | ? | - | N/A | Betula |
| MA-12 | ? | ? | ~80 | N/A | Picea |
| MA-13 | ? | ? | - | N/A | Larix |
| MA-14 | ? | ? | ~100 | N/A | Larix |
| MA-15 | ? | ? | - | N/A | Betula |
| MA-16 | ? | ? | - | N/A | Larix |
| MA-17 | ? | ? | ~160 | N/A | Larix |
| MA-18 | ? | ? | - | N/A | Alnus |
| MA-19 | 1767 | 1951 | 184 | Lena Central | Larix |
| MA-20 | 1652 | 1954 | 302 | Lena North | Larix |
| MA-21 | ? | ? | ~85 | N/A | Picea |
| MA-22 | ? | ? | - | N/A | Alnus |
| MA-23 | ? | ? | - | N/A | Betula |
| MA-24 | ? | ? | - | N/A | Alnus |
| MA-25 | ? | ? | ~80 | N/A | Picea |
| MA-34 | 1898 | 1988 | 90 | Lena North | Larix |
| MA-35 | ? | ? | - | N/A | Picea |
| MA-36 | ? | ? | - | N/A | Picea |
| MA-37 | ? | ? | ~120 | N/A | Larix |

| MA-38 | 1836 | 1988 | 152 | Lena Central; Lena North | Larix |
|---|---|---|---|---|---|

\* hyphen indicates samples omitted from analysis (less than <60 tree rings or preservation issues)


## 4 Conclusions

By comparing $^{14}$C dates from buried driftwood with dendrochronological estimates of the age of modern driftlines in an Arctic coastal setting of eastern Siberia, this study shows that reliable age indication on Holocene coastal change can be obtained from syndepositional driftwood. Conceptual age uncertainties associated with the river-borne transport of the wood appear to be insignificant for large river systems,
such as the Lena. The radiocarbon dating of driftwood has been widely used in a range of coastal settings across the Arctic, but chronologies have never before been systematically compared with the precise (dendrochronological) age of modern driftwood. The possibility to determine the age of driftwood with an annual precision may further be an ideal proxy for the study of driftwood delivery by low-frequency events (e.g. coastal storm surges). Reliable and precise chronological data are crucial for understanding
the mesoscale response of fragile coastal systems to effects of past climate variability and sea-level changes.

**Author contribution:** L.S. and U.B. designed the study, L.S. conducted the field sampling, A.K. and
A.C. carried out the analyses of the wood samples. L.S. prepared the manuscript with substantial contribution from all authors.

**Competing interests:** The authors declare that they have no conflict of interest.

**Acknowledgements:** This study received in-house funding by the Coastal Ecology section of the Alfred Wegner Institute, Helmholtz Centre for Polar and Marine Research. A.C. received funding from the Fritz and Elisabeth Schweingruber Foundation. U.B. received funding from SustES – Adaptation strategies for sustainable ecosystem services and food security under adverse environmental conditions (CZ.02.1.01/0.0/0.0/16_019/0000797), as well as from the ERC project MONOSTAR (AdG 882727).

Toru Tamura and Anthony Jull are thanked for providing valuable comments on an earlier version of this manuscript.

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
