# Peer review of "Short communication: Driftwood provides reliable chronological markers in Arctic coastal deposits"

_Geochronology, 2020_

## Referee Comment (RC1) · Anthony Jull (Referee) · 16 Nov 2020

The paper of Sander et al. looks at using the dendrodating and radiocarbon dating of driftwood in two Arctic coastal sites in Siberia. The authors are able to date past Holocene deposits using radiocarbon dates on the driftwood. Modern deposits are apparently dated only using dendrochronology. I have some minor comments.

1. In the introduction figure S1, the uncertainties in driftwood 14C ages from various other publications are presented. It seems that these data give more information that could be useful in the authors' analysis, but they just summarize them here. As the authors note about line 51, a big unknown in the assumptions of the dates presented in this paper is the duration of the time when the tree falls and the wood is transported

by a river system to the sea. Although figure S1 suggests 50-100 years is reasonable, this depends to some extent on the nature of the forest and the authors note it could be several centuries. 2. The results discussed in table 1 and figure 2 suggest that one can assign a radiocarbon date probability distribution to various discrete events. For example, BY1 and BY2 overlap well, as are MA-27, 28 and 29. A problem with figure 2 is that the images on the right are practically unreadable for the site locations – especially if one has a B&W image. I recommend these images be improved. 3. In section 3.2, no radiocarbon measurements appear to have been made on the "modern" driftwood. This seems like an important oversight. Although the authors dendrodated 4 Larix samples successfully, it would be interesting and useful to see the 14C bomb spike in the wood sequences, for example for the other species, as this would give some information also useful to the hypothesis presented.
* * *

---

## Referee Comment (RC2) · Toru Tamura (Referee) · 6 Jan 2021

The Short Communication paper by Sander et al. reports a thorough study of radiocarbon ages of drift wood buried in two arctic prograded beach-ridge plains (Bys Tasa and Makhchar) and dendrochronology of drift wood on their modern beaches. Key findings of this paper include 1) the residence time of drift wood from erosion to final deposition is typically decades long, consistent with previous estimates, and within uncertainties of state-of-art radiocarbon dating, 2) up to $\sim$300 tree rings are observed in drift wood, and 3) the radiocarbon ages of the outermost tree rings in individual samples define very consistent sequences with no age reversal in terms of the morpho-stratigraphy of the beach ridge (i.e., ridges are younger seawards). These findings are well-supported by the data and provide important insights into the radiocarbon chronology in such

settings: radiocarbon ages of drift wood, if the outermost tree rings are dated, are accurate enough to provide reliable chronology of prograded beach ridges, while if the position of the dated sample is not clear it may introduce overestimate up to a few thousand years. I agree with all the arguments in the paper and raise only one discussion point to be added (though not mandatory); at present the changes in the progradation rate (as defined in Fig. 3) are attributed to external forcing, such as wave climates and sea-ice cover that are unknown, but some changes in the Makhchar system appear to be relevant with the changing compartment size in association with the beach progradation (e.g., around 1400 BCE and 700 CE). Although further discussion should be given elsewhere, this correlation is worth mentioning here as it may provide additional confidence with the radiocarbon chronology. Minor corrections L22: transport duration or duration of transport? L37: New paragraph from 'Age control' L81: or 'have been positive over' L87: Fig. 1, Top right L122: 'and', not 'an'? Table 2: MA-4 to MA-26 may be tagged as Set 9. L141: 'conceptually' may be deleted L161-2: or 'hence minimizing reworking or erosion' L164: MA initiated around L224: crucial for understanding the mesoscale

---

## Author Comment (AC1) · 13 Jan 2021

Dear Prof. Jull, thank you very much for reading and commenting on our manuscript. Below we reply to all comments in a point-by-point fashion and repeat your question to ensure transparency for all readers interested in the public peer-review process.

**Comment 1 (Jull):** *In the introduction figure S1, the uncertainties in driftwood 14C ages from various other publications are presented. It seems that these data give more information that could be useful in the authors' analysis, but they just summarize them here.*
**Reply (Authors):** The purpose of comparing the 14C age uncertainties from different

studies here is to show that (1) the uncertainties associated with the radiocarbon-dating of driftwood are generally moderate, and (2) that the uncertainty of age determinations decreased considerably over the last 30 years (based on methodological/technological developments). We show the data to visually substantiate our argument in that regard (the main in-text reference is found in line 39-41 of the manuscript text). Discussing the data further requires insight into details (such as the calculation of individual laboratory errors), that we have discussed with our own laboratory at AWI Bremerhaven, but that are not disclosed in the reviewed studies. I hope you agree that it is useful to display the data in the supplementary file in order to build our argument: It matters to specify the position of the 14C-dated material in relation to the outermost tree-ring (e.g. regarding wood abrasion or decay, see line 206-210), as the error of not knowing likely is larger than the methodological uncertainty of the 14C method today.

**Comment 2 (Jull):** *As the authors note about line 51, a big unknown in the assumptions of the dates presented in this paper is the duration of the time when the tree falls and the wood is transported by a river system to the sea. Although figure S1 suggests 50-100 years is reasonable, this depends to some extent on the nature of the forest and the authors note it could be several centuries.*

**Reply (Authors):** In lines 53-58, we argue that the duration of river-borne transport of any individual driftwood sample of central Siberian provenance prior to deposition on an Arctic beach is and remains an unknown factor. This problem cannot be resolved in a straight-forward fashion, but as you correctly point out, the findings of our study suggest that samples from the same stratigraphic position have the same age within the uncertainty of 14C dating and calibration. We further elaborate on this matter in line 61-65 and line 200-206, and state, based on the age of the four dendro-dated samples, that transport times likely are much shorter than the uncertainty of the 14C age determination. Nevertheless, the real duration of transport remains unknown, hence we do not propose the application of a correction factor to account for time lost by transport (if this is what you are implying). The dynamics of large woody debris in

rivers still is an emerging field of research and the coming decades will shed more light on this matter – and we hope to be able to contribute to this by investigating coastal driftwood deposition.

**Comment 3 (Jull):** *The results discussed in table 1 and figure 2 suggest that one can assign a radiocarbon date probability distribution to various discrete events. For example, BY1 and BY2 overlap well, as are MA-27, 28 and 29. A problem with figure 2 is that the images on the right are practically unreadable for the site locations – especially if one has a BW image. I recommend these images be improved.*
**Reply (Authors):** The maps for the site locations in Figure 2 have been modified to improve readability (see attachment). Along with other minor adjustments, the font size was increased and numbers for the clusters were added for orientation in B/W print-outs.

**Comment 4 (Jull):** *In section 3.2, no radiocarbon measurements appear to have been made on the "modern" driftwood. This seems like an important oversight. Although the authors dendrodated 4 Larix samples successfully, it would be interesting and useful to see the 14C bomb spike in the wood sequences, for example for the other species, as this would give some information also useful to the hypothesis presented.*
**Reply (Authors):** This is right, we do rely entirely on dendro-dating for the samples from the modern beach. Your comment to use the 14C bomb spike as an independent reference (and to support the dendro-dating) is a really useful suggestion – thank you for that. For future work, we hope to be able to return to the Siberian Arctic in order to extend this study in by dating more driftwood specimen and to include the 14C dating of younger material (pre- and post-bomb).

---

## Author Comment (AC2) · 13 Jan 2021

Dear Toru, thank you very much for your constructive comments and thorough edits. We are very pleased to hear that you agree with our approach and interpretations! Please find a brief discussion of your comment regarding the changes in beach-ridge progradation rate below (see below). All minor corrections have been incorporated into the revised version of the manuscript.

**Comment (Tamura):** *I agree with all the arguments in the paper and raise only one discussion point to be added (though not mandatory); at present the changes in the progradation rate (as defined in Fig. 3) are attributed to external forcing, such as wave*

*climates and sea-ice cover that are unknown, but some changes in the Makhchar system appear to be relevant with the changing compartment size in association with the beach progradation (e.g., around 1400 BCE and 700 CE). Although further discussion should be given elsewhere, this correlation is worth mentioning here as it may provide additional confidence with the radiocarbon chronology.*

**Reply (Authors):** This is an interesting point and a very well observed detail! However, I am afraid, we are not able to respond to this in a conclusive fashion. In summer 2018, we conducted georadar surveys both in the lagoon (50Mhz) and on the beach-ridge system (250Mhz) in order to resolve information on the internal architecture of the beach deposits and to detect bedrock contacts – both with limited success, likely associated with the lithology of the beach clasts and local bedrock. In consequence, we possess no knowledge as to the deposited volumes. In the manuscript text, we only provided a progradation rate for the inner parts of the BY and MA systems, where the shape of the valley least influenced the availability of accommodation space. At the point in time you mention above, the compartment size at MA changes dramatically, which must have influenced deposition and, as you suggest, may have prematurely reduced progradation rates. This landscape feedback locally masks the effect of climate and sediment supply, and is an unfortunate coincidence and a good example of a limitation in beach-ridge research. Buor Khaya Bay features at least three more beach-ridge sites, that are worth investigating and we hope that, by future studies, these local effects will eventually even out. With this present study, we have the chronological toolbox in place to be able to potentially resolve these details. We hope to be able to return to the area in the near future and would be happy to discuss the results with the community. Thank you very much for your comment!

---

## Author Response (AR1)

Dear Philippa and Irka,

Thank you very much for your positive response and your editorial work!

I have now uploaded the final revised version of our manuscript, both as a .pdf- and .docx-file. As you encouraged, this version includes an additional sentence on the potential to trace spikes in the atmospheric radiocarbon concentration in order to date driftwood (line 213-216). Please be aware, that a few additional changes of purely technical nature have been made in the text (most of which suggested by R2). I would be happy to supply a document with track-changes to make this transparent. All figures have been modified to improve clarity and readability. These new versions are included in the above documents and separately attached as 300 dpi .jpg-files in a single .zip-archive.

Please let me know if any further changes are necessary.

With kind regards,
Lasse
(on behalf of all authors)